# Building Instruction-Tuning Datasets from Human-Written Instructions with Open-Weight Large Language Models

**Youmi Ma[1]   Sakae Mizuki[1,2]   Kazuki Fujii[1,2]   Taishi Nakamura[1,2]**
**Masanari Ohi[1,2]   Hinari Shimada[1]   Taihei Shiotani[1]   Koshiro Saito[1]**
**Koki Maeda[1]   Kakeru Hattori[1,2]   Takumi Okamoto[1]   Shigeki Ishida[1]**
**Rio Yokota[1,2,3]   Hiroya Takamura[2]   Naoaki Okazaki[1,2,3]**

[1] Department of Computer Science, School of Computing, Institute of Science Tokyo
[2] National Institute of Advanced Industrial Science and Technology
[3] Research and Development Center for Large Language Models, NII
{ma.y@, swallow@nlp., okazaki@}comp.isct.ac.jp

## Abstract

Instruction tuning is crucial for enabling Large Language Models (LLMs) to solve real-world tasks. Prior work has shown the effectiveness of instruction-tuning data synthesized solely from LLMs, raising a fundamental question: Do we still need human-originated signals for instruction tuning? This work answers the question affirmatively: we build state-of-the-art instruction-tuning datasets sourced from human-written instructions, by simply pairing them with LLM-generated responses. LLMs fine-tuned on our datasets consistently outperform those fine-tuned on existing ones. Our data construction approach can be easily adapted to other languages; we build datasets for Japanese and confirm that LLMs tuned with our data reach state-of-the-art performance. Analyses suggest that instruction-tuning in a new language allows LLMs to follow instructions, while the tuned models exhibit a notable lack of culture-specific knowledge in that language. The datasets and fine-tuned models will be publicly available[1]. Our datasets, synthesized with open-weight LLMs, are openly distributed under permissive licenses, allowing for diverse use cases.

## 1 Introduction

Open-weight Large Language Models (LLMs) have seen rapid advancements over the past year. Models like Llama-3.1 (Grattafiori et al., 2024) and Qwen-2.5 (Qwen et al., 2024) now achieve performance comparable to proprietary models such as GPT-4 (OpenAI et al., 2024) and Gemini (Google et al., 2024b). A crucial technology behind this is instruction-tuning (Wei et al., 2022), a supervised fine-tuning process that equips models with abilities to engage in conversations and generate responses that satisfy human needs.

Unfortunately, even for LLMs whose weights are open, data used for instruction tuning are mostly closed. The data, typically consisting of (*instruction*, *response*) pairs, are keys to democratizing LLM: it remains an open question what kind of instructions are effective in enhancing LLMs' instruction-following abilities. Xu et al. (2025) proposed to "extract" the instructions directly from the instruction-tuned models by prompting with a pre-query template. Datasets constructed through this process, named Magpie, have demonstrated remarkable effectiveness, being state-of-the-art among all publicly available instruction-tuning datasets. The success of the self-synthesized data raises an important question: Do we still need human-originated signals for instruction tuning? In other words, are LLM-generated instructions truly superior to human-written ones?

This work seeks an answer to the above question. To this end, we synthesize datasets from human-written instructions, as in Figure 1. For instructions, we extract user utterances

---

[1] https://huggingface.co/datasets/tokyotech-llm/lmsys-chat-1m-synth

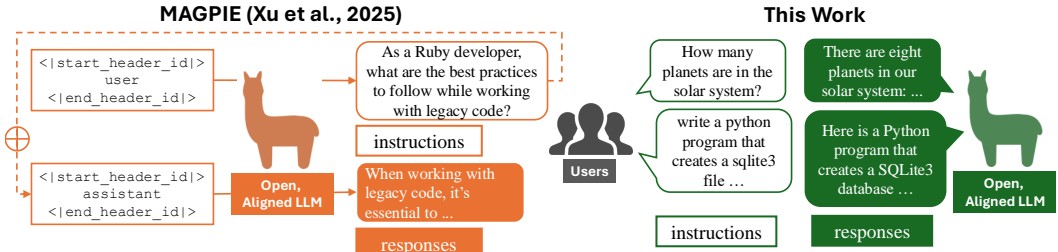

Figure 1: Overview of how Magpie (Xu et al., 2025), the existing state-of-the-art dataset and our dataset are collected. Starting from instructions written by humans, we synthesize responses by open-weight LLMs.

| Dataset | Instruction | Response | Language | # Instances |
|---|---|---|---|---|
| Dolly (Conover et al., 2023) | Human | Human | en. | 15,011 |
| OpenAssistant2 (Köpf et al., 2023) | Human | Human | en. | 128,575 |
| OpenOrca (Lian et al., 2023) | Human | GPT-3.5, GPT-4 | en. | 4,233,923 |
| LMSYS-Chat-1M (Zheng et al., 2024) | Human | 25 LLMs | en. | 1,000,000 |
| WildChat (Zhao et al., 2024) | Human | GPT-3.5, GPT-4 | en. | 529,428 |
| Alpaca (Taori et al., 2023) | GPT-3.5 | GPT-3.5 | en. | 52,002 |
| Baize (Xu et al., 2023) | GPT-3.5 | GPT-3.5 | en. | 210,311 |
| UltraChat (Ding et al., 2023) | GPT-3.5 | GPT-3.5 | en., zh. | 1,468,352 |
| Evol-Instruct (Xu et al., 2024) | GPT-3.5 | GPT-3.5 | en. | 70,000 |
| Magpie-Ultra-v0.1 (Xu et al., 2025) | Llama-3.1-405B-Instruct | Llama-3.1-405B-Instruct | en. | 49,052 |
| Magpie-Pro-Filtered (Xu et al., 2025) | Llama-3.1-70B-Instruct | Llama-3.1-70B-Instruct | en. | 300,000 |
| Ja-Self-Instruct (Sun et al., 2024) | GPT-4 | GPT-4 | ja. | 52,002 |
| **This work** | | | | |
| Proposed-Llama-3.1-En | Human | Llama-3.1-405B-Instruct | en. | 453,055 |
| Proposed-Gemma-2-En | Human | Gemma-2-27B-IT | en. | 453,671 |
| Proposed-Llama-3.1-Ja | Human | Llama-3.1-405B-Instruct | ja. | 453,802 |
| Proposed-Gemma-2-Ja | Human | Gemma-2-27B-IT | ja. | 451,450 |

Table 1: Statistics of existing and our datasets. The column **Language** shows the major language in each dataset. We are the first to publish datasets with human-written instructions and responses synthesized by open-weight LLMs.

from LMSYS-Chat-1M (Zheng et al., 2024), a dataset composed of real-world conversations between humans and chatbots. For responses, we synthesize assistant utterances using open-weight LLMs, namely Llama-3.1 (Grattafiori et al., 2024) and Gemma-2 (Google et al., 2024a) utilized for synthesizing Magpie (Xu et al., 2025). We fine-tuned four leading base LLMs on our datasets and compared their performance against those fine-tuned on Magpie. Three out of four models demonstrated superiority with our dataset, while the remaining model performed comparably. The results highlight the superiority of human-written instructions compared to LLM-generated ones. Pairing human-written instructions with LLM-generated responses thus leverages the strengths of both humans and LLMs, resulting in state-of-the-art datasets for instruction tuning.

A practical issue of our proposed method could be the difficulty of collecting large-scale human-written instructions, since existing resources are limited to English. To this end, we verify if our method is still effective in non-English languages based on machine translation. Specifically, we translated English human-written instructions into the target language and then prompted LLMs to synthesize responses. We chose Japanese as the target language and synthesized two datasets. Fine-tuning with the datasets significantly boosted the instruction-following abilities of all five base LLMs with different levels of Japanese proficiency. In total, this work built four datasets, with statistics shown in Table 1. We further analyzed how the instruction-following abilities transfer between languages. From the analyses, LLMs trained with Japanese datasets can solve surface-level editing (creative writing and roleplay) and region-agnostic knowledge domain (STEM) but are worse at instructions that require culture-specific knowledge.

In short, the contributions of this work are: (1) We construct and publish state-of-the-art instruction-tuning datasets with permissive licenses, allowing for a wide range of use cases. Fine-tuning LLMs with our datasets yields the best performance among all publicly available

datasets. (2) We highlight the importance of human-written instructions collected from real-world human-chatbot interactions. By leveraging both human- and LLM-originated signals we obtain datasets with quality higher than those solely synthesized from LLMs. (3) We demonstrate that instruction-tuning in a new language equips LLMs with abilities to follow instructions, but it does not improve culture-specific knowledge.

## 2 Synthesized Data Generation

Starting from human-written instructions, we synthesize responses with open-weight LLMs to construct instruction-tuning datasets. Here we note the dataset as $\mathcal{D} = \{(I_1, R_1), (I_2, R_2), \cdots, (I_n, R_n)\}$, where $I_k$ and $R_k$ ($k \in \{1, 2, \cdots, n\}$) represent an instruction and its corresponding response.

**Instructions.** We collected instructions $\mathcal{I} = \{I_1, I_2, \cdots, I_n\}$ from LMSYS-Chat-1M (Zheng et al., 2024)[2], a dataset of one million real-world conversation records between human and LLMs. The dataset was selected as our foundation as it covers a wide range of topics, with a permissive license for both academic and commercial use. Notably, some of the conversations in LMSYS-Chat-1M are flagged as toxic; these conversations were excluded as our focus here is to improve LLMs' instruction-following abilities, i.e., how accurately LLMs can fulfill user requests. After filtering out toxic conversations, we obtained 732,392 instances, from which the first user utterance is selected as the instruction. Further removal of duplications left us with 453,889 instructions directly adopted for synthesizing responses.

**Responses.** Instances in LMSYS-Chat-1M include model responses. Nevertheless, we excluded these responses from our dataset because they were collected from a mixture of models, including both proprietary and open-weight LLMs. To ensure the quality of responses and a permissive license, we employed a leading open-weight LLM $\hat{\pi}$ to generate response $R_k$ corresponding to each instruction $I_k$. This work synthesized two datasets from the same instruction set, utilizing different LLMs to generate responses: Llama-3.1-405B-Instruct and Gemma-2-27B-IT. Responses from Llama-3.1-405B-Instruct were generated via the API provided by DeepInfra[3], while responses from Gemma-2-27B-IT were generated using vLLM (Kwon et al., 2023) on a single NVIDIA H100 SXM5 GPU. Responses with over 2,000 tokens were considered repetitions and were evicted from the final dataset.

Statistics of our datasets, {Llama-3.1 | Gemma-2}-Proposed-En, are reported in Table 1. A dataset $D$ synthesized by an LLM $\hat{\pi}$ enables models to imitate how $\hat{\pi}$ responds to human-written instructions. We hereafter refer to the LLM utilized for response generation as a **teacher model** and the LLM fine-tuned on the dataset as a **student model**.

## 3 Experiments

We validate if LLMs instruction-tuned on our datasets outperformed their counterpart tuned on the Magpie (Xu et al., 2025) family. Notably, the dataset family has been recognized as state-of-the-art among public instruction-tuning datasets. We expect the results to explain whether human-written instructions are still necessary for instruction tuning.

### 3.1 Configurations

**Student Models.** We adopted popular open-weight LLMs as our test bed. We experimented on Qwen-2.5-7B (Qwen et al., 2024), Llama-3.1-8B (Grattafiori et al., 2024), Gemma-2-9B (Google et al., 2024a), and OLMo-2-1124-13B (OLMo et al., 2024). We used pre-trained models from Huggingface Hub.

---

[2] https://huggingface.co/datasets/lmsys/lmsys-chat-1m
[3] https://deepinfra.com/meta-llama/Meta-Llama-3.1-405B-Instruct

| Base Model | SFT Dataset | MT-Bench | | |
| --- | --- | --- | --- | --- |
| | | Average | 1st Turn | 2nd Turn |
| *Reference: performance of teacher models* | | | | |
| Llama-3.1-405B-Instruct | | $8.33_{\pm.08}$ | $8.39_{\pm.08}$ | $8.27_{\pm.12}$ |
| Gemma-2-27B-IT | | $7.89_{\pm.05}$ | $8.19_{\pm.06}$ | $7.54_{\pm.10}$ |
| Llama-3.1-8B | LMSYS-Chat-1M-Random | $4.10_{\pm.08}$ | $4.77_{\pm.15}$ | $3.36_{\pm.14}$ |
| | LMSYS-Chat-1M-Best | $4.15_{\pm.08}$ | $4.89_{\pm.05}$ | $3.36_{\pm.13}$ |
| | Ultra-Chat | $5.13_{\pm.11}$ | $5.76_{\pm.05}$ | $4.43_{\pm.18}$ |
| | Magpie-Ultra-v0.1 | $5.61_{\pm.11}$ | $6.32_{\pm.06}$ | $4.84_{\pm.27}$ |
| | Magpie-Gemma2-Pro-Filtered | $5.82_{\pm.12}$ | $6.30_{\pm.07}$ | $5.24_{\pm.21}$ |
| | Magpie-Llama-3.1-Pro-MT-Filtered | $6.40_{\pm.08}$ | $6.86_{\pm.11}$ | $5.90_{\pm.16}$ |
| | Proposed-Llama-3.1-En | $\mathbf{6.82}_{\pm.08}$ | $\mathbf{7.54}_{\pm.08}$ | $6.02_{\pm.15}$ |
| | Proposed-Gemma-2-En | $6.40_{\pm.10}$ | $6.58_{\pm.06}$ | $\mathbf{6.21}_{\pm.17}$ |
| Qwen-2.5-7B | Magpie-Llama-3.1-Pro-MT-Filtered | $5.61_{\pm.05}$ | $6.33_{\pm.07}$ | $4.82_{\pm.07}$ |
| | Proposed-Llama-3.1-En | $\mathbf{7.10}_{\pm.12}$ | $\mathbf{7.67}_{\pm.06}$ | $\mathbf{6.47}_{\pm.21}$ |
| | Proposed-Gemma-2-En | $6.92_{\pm.06}$ | $7.59_{\pm.07}$ | $6.17_{\pm.13}$ |
| Gemma-2-9B | Magpie-Llama-3.1-Pro-MT-Filtered | $6.47_{\pm.08}$ | $7.23_{\pm.03}$ | $5.65_{\pm.16}$ |
| | Proposed-Llama-3.1-En | $\mathbf{6.79}_{\pm.06}$ | $\mathbf{7.39}_{\pm.07}$ | $\mathbf{6.13}_{\pm.06}$ |
| | Proposed-Gemma-2-En | $6.58_{\pm.06}$ | $7.11_{\pm.03}$ | $5.97_{\pm.12}$ |
| OLMo-2-1124-13B | Magpie-Llama-3.1-Pro-MT-Filtered | $\mathbf{6.85}_{\pm.10}$ | $7.16_{\pm.02}$ | $\mathbf{6.50}_{\pm.21}$ |
| | Tülu 3[†] | $5.92_{\pm.16}$ | $6.41_{\pm.13}$ | $5.38_{\pm.22}$ |
| | Proposed-Llama-3.1-En | $6.49_{\pm.08}$ | $\mathbf{7.50}_{\pm.04}$ | $5.39_{\pm.18}$ |
| | Proposed-Gemma-2-En | $5.44_{\pm.12}$ | $6.95_{\pm.11}$ | $3.75_{\pm.21}$ |

Table 2: Model performance evaluated on MT-Bench. † represents the official recipe utilized in the supervised fine-tuning of OLMo-2-1124-13B, with evaluation results of `allenai/OLMo-2-1124-13B-SFT` reported. 3 out of 4 LLMs fine-tuned on our datasets outperform those fine-tuned on Magpie, the previous state-of-the-art dataset family.

**Training.** We conducted full-parameter Supervised Fine-Tuning (SFT) for every instruction-tuning experiment reported in this paper. Specifically, all model parameters were updated to maximize the log probability of response $R_k$ given instruction $I_k$. The training process utilizes Huggingface's transformers library (Wolf et al., 2020), and distributed across 4 or 8 NVIDIA H100 SXM5 GPUs using DeepSpeed ZeRO (Rajbhandari et al., 2020). All experiments were finished in 24 hours except those with Gemma-2-9B, which took 40 hours at maximum. AdamW (Loshchilov & Hutter, 2019) was adopted as the optimizer, with $\beta_1 = 0.9$ and $\beta_2 = 0.95$. The learning rate was adjusted following a cosine annealing schedule, which linearly increased to the peak at $2.5 \times 10^{-5}$ in the first 10% number of steps, then smoothly decreased to $2.5 \times 10^{-6}$. Each training process ran for 2 epochs with batch size 512 across all experiments.

**Evaluation.** The performance of trained models was evaluated on Multi-Turn Bench (MT-Bench), a benchmark known for high agreement with human preferences (Zheng et al., 2023)[4]. MT-Bench consists of 80 high-quality multi-turn data spanning eight categories. MT-Bench follows an LLM-as-a-Judge scheme, utilizing GPT-4 to score model responses from 1 to 10. We adopted *gpt-4-1106-preview* as the judge for all experiments. For each instruction in MT-Bench, we sampled five responses from each model, scored each response with the judge, and reported the average score and standard derivation of five runs.

### 3.2 Main Results

This subsection explores how our synthesized datasets improve LLMs' abilities to follow instructions. We compare the performance of models fine-tuned using our datasets with **two groups of datasets**: baselines and strong competitors.

**Baselines.** Comparison with baselines serves as a sanity check. To justify our efforts in synthesizing additional responses, it is necessary to show that LLMs fine-tuned on

---

[4]https://github.com/lm-sys/FastChat

our datasets should outperform those fine-tuned on the baseline datasets. We included two baselines derived from the same dataset as ours, i.e., LMSYS-Chat-1M, by using the instruction-response pairs in their original form. For instructions with multiple responses, we either randomly picked one (noted as LMSYS-Chat-1M-Random) or chose the best one judged by Gemma-2-27B-IT (noted as LMSYS-Chat-1M-Best).

**Strong Competitors.** Comparison with existing datasets provides direct insights into answering the research question. Our major competitor is Magpie (Xu et al., 2025), reporting state-of-the-art performance among all publicly available instruction-tuning datasets. Here we include three variants: Magpie-Gemma2-Pro-Filtered, Magpie-Llama-3.1-Pro-MT-Filtered, and Magpie-Ultra-v0.1 synthesized with Gemma-2-27B-IT, Llama-3.1-70B-Instruct, and Llama-3.1-405B-Instruct, respectively. Additionally, we include UltraChat (Ding et al., 2023) as the representative of datasets synthesized with proprietary LLMs.

The results are presented in Table 2. We detail the results of Llama-3.1-8B tuned on each baseline, strong competitor, and our dataset. For other student models, only the strongest competitor, i.e., Magpie-Llama-3.1-Pro-MT-Filtered is included.

Firstly, compared to the baselines, Llama-3.1-8B fine-tuned on Proposed-Llama-3.1-En leads its counterpart fine-tuned on LMSYS-Chat-1M-Best by over 2.5 points. Notably, the baselines include responses generated by proprietary LLMs such as the GPT and Claude families. Despite this, fine-tuning on baselines still underperforms ours. Responses in LMSYS-Chat-1M were collected in mid-2023, nearly one year before the release of the teacher models we adopted. The results show that thanks to the rapid advancements in open-weight LLMs over the past year, updating assistant responses with cutting-edge LLMs effectively enhances the utilization of LMSYS-Chat-1M.

Next, compared to existing instruction-tuning datasets, Llama-3.1-8B fine-tuned on Proposed-Llama-3.1-En shows clear advantages. Compared to UltraChat, which is synthesized using GPT-4, our datasets exhibit better performance across all cases. Against the Magpie family synthesized using open-weight LLMs, i.e., the state-of-the-art datasets for instruction tuning, the performance benefits are also observable. Note that Proposed-Gemma-2-En and Magpie-Gemma2-Pro-Filtered, Proposed-Llama-3.1-En and Magpie-Ultra-v0.1 share the same LLM for synthesized data generation. The difference is instructions: while Magpie datasets employ LLM-generated instructions, our datasets utilize human-written ones. The superiority of our datasets, hence, stems from human-written instructions, answering our research question: **human-written instructions remain useful for instruction-tuning**. Our approach is thus promising for building instruction-tuning datasets by leveraging both human and machine resources.

The superiority of our datasets can also be observed when fine-tuning Gemma-2-9B and Qwen-2.5-7B. For OLMo-2-1124-13B, our datasets slightly lag behind Magpie-Llama-3.1-Pro-MT-Filtered in overall performance but outperform it in the first turn. Notably, **Magpie-Llama-3.1-Pro-MT-Filtered is a multi-turn dataset, while ours are single-turn**. The superiority in first-turn performance can thus be considered sufficient validation of the effectiveness of our datasets. In addition, OLMo-2-1124-13B fine-tuned on our dataset surpasses the performance of the official model after SFT. As reported in OLMo et al. (2024), the recipe for SFT is a mixture of recent datasets such as WildChat (Zhao et al., 2024) synthesized by GPT-4. Nevertheless, the performance is lower than OLMo-2-1124-13B fine-tuned on Proposed-Llama-3.1-En, suggesting the effectiveness of our datasets.

### 3.3 Instance-Wise Effectiveness

LLMs fine-tuned on our datasets outperform those tuned on existing ones. However, the scales of datasets differ (Table 1): our datasets contain approximately 100K more instances than Magpie-Llama-3.1-Pro-MT-Filtered (Xu et al., 2025). Here, we decouple the effect from the dataset scale and the quality of each instance by aligning the number of training instances to that of the competitors.

| SFT Data | # Instances | MT-Bench |
|---|---|---|
| Magpie-Llama-3.1-Pro | 300,000 | 6.40 |
| Proposed-Llama-3.1-En | 300,000 | **6.54** |
| Magpie-Ultra-v0.1 | 49,052 | 5.61 |
| Proposed-Llama-3.1-En | 49,052 | **6.20** |
| Magpie-Gemma2-Pro | 200,000 | 5.82 |
| Proposed-Gemma-2-En | 200,000 | **6.50** |

Table 3: Performance of Llama-3.1-8B evaluated on MT-Bench when the number of training instances is aligned. Fine-tuning on our dataset, even downsampled, outperforms those fine-tuned on the magpie family.

**Settings.** We downsampled (1) Proposed-Llama-3.1-En to compare with Magpie-Llama-3.1-Pro-MT-Filtered; (2) Proposed-Llama-3.1-En to compare with Magpie-Ultra; and (3) Proposed-Gemma-2-En to compare with Magpie-Gemma-2-Pro-Filtered. For downsampling, we followed the same filtering strategy as Xu et al. (2025) to assign a quality score to each synthesized response with LLM, then select the top-N (*instruction*, *response*) instances for instruction tuning. For Llama-based datasets, we adopted Llama-3.1-70B-Instruct for automatic scoring and selected the top 300K instances; for Gemma-based datasets, we adopted Gemma-2-27B-IT for scoring and selected the top 200K instances. The results on Llama-3.1-8B are reported in Table 3.

**Results.** Even when the number of training instances is controlled, Llama-3.1-8B fine-tuned with our datasets still outperforms those tuned with Magpie. The performance gap is consistent for all settings, verifying the instance-wise effectiveness of our datasets. Notably, the dataset pairs in the last two groups utilize the same LLM for generating responses, and the number of instances is controlled to be the same: The performance gap thus arises solely from differences in the instructions. This further validates the effectiveness of human-written instructions in LMSYS-Chat-1M.

## 4 Synthesizing Data in Other Languages

Section 3 witnessed the effectiveness of our datasets in instruction tuning. The data construction approach, however, is adaptable for synthesizing datasets in other languages. This section examines whether this approach can generate high-quality datasets for non-English instruction-tuning, using Japanese as the representative.

To construct the instruction-tuning dataset, we translated the instruction set $\mathcal{I}$ from Section 2 into Japanese with the DeepL translator API[5]. Given the translated instructions, Llama-3.1-405B-Instruct and Gemma-2-27B-IT were utilized for response generation. This process yields {Llama-3.1 | Gemma-2}-Proposed-Ja in Table 1.

### 4.1 Configurations

The configurations are identical to those in Section 3.1, except for the following differences.

**Student Models.** OLMo-2-1124-13B is excluded due to its limited proficiency in Japanese. Instead, we added Llama-3.1-Swallow-8B (Fujii et al., 2024; Okazaki et al., 2024) and llm-jp-3-13b (Aizawa et al., 2024), where the former undergoes continuous pre-training from Llama-3.1-8B on a Japanese corpus, and the latter is trained from scratch with a special focus on Japanese capability.

**Evaluation.** We adopt Japanese MT-Bench, a variant of MT-Bench that reflects Japan's language, education, society, and culture[6]. Japanese is one of few languages with reliable

---

[5]https://www.deepl.com/en/translator
[6]https://wandb.ai/wandb-japan/llm-leaderboard/reports/Nejumi-LLM-Neo--Vmlldzo2MTkyMTU0

| Model | SFT Data | MT-Bench | Ja. char. ratio |
|---|---|---|---|
| *Reference: performance of teacher models* | | | |
| Gemma-2-27B-IT | – | $7.10_{\pm.06}$ | 0.66 |
| Llama-3.1-405B-Instruct | – | $6.79_{\pm.17}$ | 0.67 |
| Llama-3.1-8B | official | $4.65_{\pm.15}$ | 0.65 |
| | Ja-Self-Inst | $3.85_{\pm.15}$ | 0.77 |
| | Proposed-Gemma-2-Ja | $\mathbf{5.64}_{\pm.07}$ | 0.65 |
| | Proposed-Llama-3.1-Ja | $4.42_{\pm.16}$ | 0.66 |
| Qwen-2.5-7B | official | $6.13_{\pm.10}$ | 0.82 |
| | Ja-Self-Inst | $4.24_{\pm.15}$ | 0.73 |
| | Proposed-Gemma-2-Ja | $\mathbf{6.39}_{\pm.09}$ | 0.67 |
| | Proposed-Llama-3.1-Ja | $5.40_{\pm.09}$ | 0.66 |
| Llama-3.1-Swallow-8B | official | $5.33_{\pm.07}$ | 0.68 |
| | Ja-Self-Inst | $4.53_{\pm.08}$ | 0.73 |
| | Proposed-Gemma-2-Ja | $\mathbf{6.23}_{\pm.12}$ | 0.65 |
| | Proposed-Llama-3.1-Ja | $5.18_{\pm.05}$ | 0.69 |
| Gemma-2-9B | official | $\mathbf{6.75}_{\pm.10}$ | 0.66 |
| | Ja-Self-Inst | $3.98_{\pm.15}$ | 0.72 |
| | Proposed-Gemma-2-Ja | $6.49_{\pm.17}$ | 0.66 |
| | Proposed-Llama-3.1-Ja | $5.14_{\pm.16}$ | 0.67 |
| llm-jp-3-13b | official | $5.18_{\pm.08}$ | 0.69 |
| | Ja-Self-Inst | $3.71_{\pm.06}$ | 0.73 |
| | Proposed-Gemma-2-Ja | $\mathbf{5.38}_{\pm.07}$ | 0.64 |
| | Proposed-Llama-3.1-Ja | $4.74_{\pm.21}$ | 0.71 |

Table 4: Model performance evaluated on Japanese MT-Bench. 4 out of 5 LLMs show better performance when fine-tuned on our dataset than their official post-trained release.

evaluation baselines, which motivated our choice. In addition, we measure the **Japanese character ratio** (**Ja. char. ratio** in Table 4) in the generated outputs to assess the input-output language consistency. A low Japanese character ratio indicates that the model generates output in off-target languages (Sennrich et al., 2024). During pilot experiments, we empirically found that a Japanese character ratio below 0.60 indicates a noticeable presence of content written in off-target languages.

## 4.2 Results

**Competitors.** Unlike English, where extensive research has been conducted on instruction tuning, Japanese remains relatively underexplored in this area, with Ja-Self-Instruct being the most recent research introducing a dataset synthesized with GPT-4 (Sun et al., 2024). We thus compare the performance of LLMs fine-tuned on our datasets with (1) their counterparts fine-tuned on Ja-Self-Instruct; and (2) their official post-trained versions. Experimental results are presented in Table 4, including the performance of GPT-3.5, GPT-4, GPT-4o, and the teacher models for reference.

Firstly, models fine-tuned with our datasets outperform those fine-tuned with Ja-Self-Instruct. As Ja-Self-Instruct is synthesized with GPT-4 (*gpt-4-0613*), a proprietary LLM, this highlights the advancement of open-weight LLMs in the past year. Notably, when compared to their officially post-trained counterparts, models tuned on our datasets still exhibit comparable or better performance. Specifically, Gemma-2-9B tuned on our dataset slightly underperforms the official Gemma-2-9B-IT, whereas all other models surpass their official counterparts. Therefore, **the proposed approach of combining human-written instructions with LLM-generated responses for instruction tuning is effective for non-English languages as well**, with instructions translated from English into the target language.

Next, we observe that the dataset synthesized with Gemma-2-27B-IT functions much better than that synthesized with Llama-3.1-405B-Instruct. While Gemma-2-27B-IT merely outperforms Llama-3.1-405B-instruct by 0.32 points, the gap between student models fine-tuned on the corresponding synthesized datasets is larger. This is counterintuitive, as we have witnessed the performance of the student model being proportional to that of the teacher

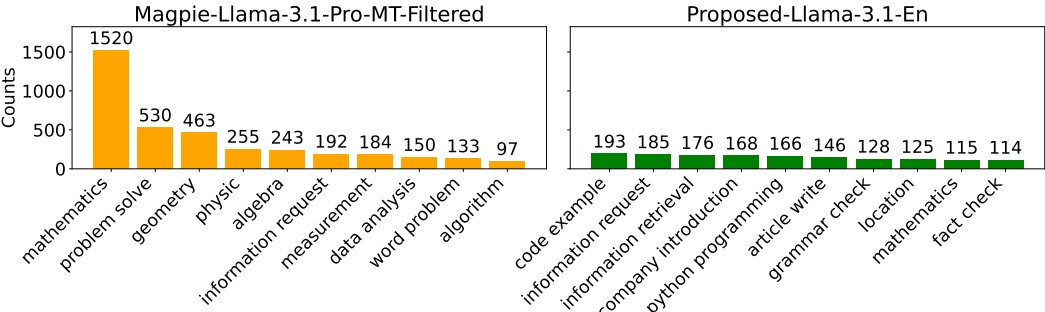

Figure 2: Counts of top-10 most frequent topics in Magpie-Llama-3.1-Pro-MT-Filtered (Xu et al., 2025) and Proposed-Llama-3.1-En. Topics in Magpie are skewed to mathematics, while those in the proposed dataset are more balanced.

model in English. This indicates that the effectiveness of model imitation for non-English languages does not necessarily align with the findings in English.

In terms of the Japanese character ratio, models trained on our datasets consistently maintain a safe level above 0.60, matching that of their teacher models. Two native-level Japanese speakers with PhD degrees reviewed the responses generated by the teacher models and found off-target language in fewer than 0.5% of cases. Given that models fine-tuned on our datasets exhibit a similar Japanese character ratio as their corresponding teacher models, we conclude that the risk of these models producing responses in off-target languages is low.

## 5 Analysis

### 5.1 Quantitative Analysis

We have shown that human-written instructions in our proposed dataset are superior to the LLM-generated instructions in Magpie (Xu et al., 2025). To better understand their difference, we dive into the instructions in Magpie-Llama-3.1-Pro-MT-Filtered and Proposed-Llama-3.1-En. We use InsTag (Lu et al., 2024) to annotate each user instruction with its semantic intent, referred to as the **topic**, to examine differences between Magpie and our dataset.

Figure 2 presents the top-10 most frequent topics and their counts. In Magpie, we observe a highly skewed distribution: the most frequent topic, namely *mathematics*, appears 1,520 times—nearly three times more than the second most frequent topic (530 times). In contrast, our proposed dataset exhibits a more uniform distribution across the top-10 topics, covering diverse categories such as article writing, grammar checking, and general reasoning. This analysis reveals that Magpie's instruction set is heavily biased toward mathematical topics, whereas our dataset provides broader and more balanced topic coverage. Such diversity likely contributes to the superior performance of models fine-tuned on our data. We also observe a correlation between the category-wise performance of fine-tuned models and the topic distributions quantified in this section. Further details are provided in Appendix E.

### 5.2 Cross-Lingual Analysis

Our datasets enable cross-lingual analyses of instruction tuning. Here, we analyze how language capabilities are enhanced after instruction-tuning (IT) and distinguish them from those enhanced after continuous pre-training (CPT). Pretraining LLMs per language from scratch demands extensive data and computation resources (Zeng et al., 2023; Aizawa et al., 2024); to avoid this, research has been conducted to adapt LLMs to underrepresented languages via IT (Chen et al., 2024; Shaham et al., 2024; Zhang et al., 2024). Concurrent to this work, Wang et al. (2025) have reported that incorporating CPT before IT better improves the capability in the target language. However, the role of CPT and IT in enhancing language capabilities remains unclear. We decouple the language capabilities into two categories,

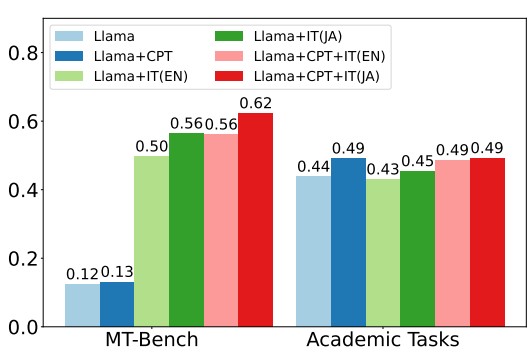
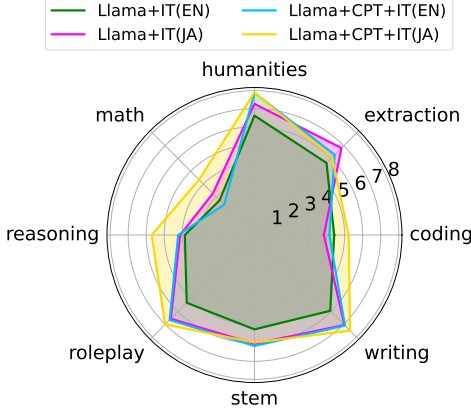

(a) Scores of MT-Bench are normalized to 0–1. IT(JA) enhances performance on MT-Bench but does not help academic tasks.

(b) IT helps `writing`, `stem`, and `roleplay`, while CPT contributes to `humanities`.

Figure 3: Model performance on each sub-category of Japanese MT-Bench.

namely **language understanding** and **language utilization**, and investigate how CPT and IT contribute to acquiring these capabilities.

**Settings.** We fine-tuned Llama-3.1-8B and Llama-3.1-Swallow-8B-v0.1 with the Gemma-2 versions of the synthesized datasets. Compared to Wang et al. (2025), which employs LlaMA-2 as the test bed, our setup is strong: Llama-3.1-Swallow-8B is the state-of-the-art Japanese LLM obtained through CPT from Llama-3.1-8B, and our datasets are the state-of-the-art for IT. For evaluation, we adopted the metrics from Fujii et al. (2024) to assess LLMs' proficiency in language understanding[7], and used the Japanese MT-Bench to evaluate their ability in language utilization. The results are presented in Figure 3a.

**Language Understanding.** On academic tasks, we observe significant performance improvement in models that underwent CPT. In contrast, models without CPT show only subtle changes in performance, even after IT with the Japanese dataset. These results suggest that CPT is crucial in equipping LLMs with fundamental language understanding abilities.

**Language Utilization.** On MT-Bench, Llama-3.1-Swallow fine-tuned on the Japanese dataset (Llama+CPT+IT(JA)) scores the highest. Interestingly, Llama-3.1 fine-tuned on the Japanese dataset (Llama+IT(JA)) and Llama-3.1-Swallow fine-tuned on the English dataset (Llama+CPT+IT(EN)) achieves comparable performance on MT-Bench, while the former underperforms the latter on the academic task. The result indicates a lack of factual knowledge about the Japanese language in Llama-3.1; nevertheless, the model follows Japanese instructions as well as Llama-3.1-Swallow fine-tuned on the English dataset.

**Task-Wise Performance.** The performance of fine-tuned models on each sub-category of the Japanese MT-Bench as in Figure 3b. The figure portrays a clear contribution of CPT in the `humanities` category, which includes topics about Japanese history, finance, and social manners. This indicates that following instructions related to cultural-specific topics requires CPT. The findings correspond to those reported in Romanou et al. (2025), saying that multilingual LLMs struggle with cultural questions, especially in languages not present in pre-training. Meanwhile, for categories that rely on global knowledge already present in the pre-trained LLM (i.e., `stem`) or those categories that require no factual knowledge (i.e., `roleplay`, `writing`, and `extraction`), IT works sufficiently well on its own. Capabilities to follow instructions in such categories are likely transferrable across languages.

---

[7]The benchmark covers a wide range of tasks such as Japanese question-answering, summarization, and translation. We only report the average across all tasks.

## 6 Related Work

Instruction-tuning datasets are typically sourced from three origins: humans, human-LLM interactions, and LLMs (Zhang et al., 2023).

**Humans.** Wei et al. (2022) introduced the first instruction tuning dataset as a collection of academic tasks in NLP research. They constructed a dataset from existing datasets, covering tasks such as question answering, translation, and summarization. These language resources are of high-quality, but do not represent the real-world use cases of current AI assistants. Recently, efforts have been made to construct instruction-tuning datasets that better reflect real-world needs. Köpf et al. (2023) recruited crowd-source workers worldwide to collect a conversation corpus simulating human-assistant conversations. To ensure the expertise of annotators, Conover et al. (2023) employed experts working in the data science domain. These datasets face challenges as being simulations, resulting in a gap between constructed data and real-world needs.

**Human-LLM Interactions.** To build datasets that fit into real-world settings, efforts have been made to gather chatlogs between humans and AI assistants in the wild. Zheng et al. (2024) collects human-AI conversations from a web service called Chatbot Arena with 25 LLMs. Zhao et al. (2024) is a concurrent work with the AI assistant restricted to the GPT family. While these datasets consist of diverse user instructions, the quality of assistant response could not be ensured. Moreover, responses generated from proprietary LLMs could not be utilized for LLM development due to the restrictiveness, in particular around the licensing of model outputs. This work values the diversity of user instructions in chatlogs by pairing them with responses generated from state-of-the-art open-weight LLMs.

**LLMs.** The community has been focusing on automatically constructing data with proprietary LLMs. LLMs can generate instruction-tuning datasets based on a minimum set of high-quality human-written (*instruction*, *response*) pairs (Taori et al., 2023; Wang et al., 2023; Sun et al., 2024). The generated dataset can be improved by utilizing LLMs to rewrite the instructions (Xu et al., 2024) or utilizing multiple LLMs to chat with each other (Ding et al., 2023; Xu et al., 2023). However, synthesized with proprietary LLMs, the usage of these datasets is restricted due to license issues. More recently, Xu et al. (2025) synthesized datasets using open-weight LLMs by generating completions from system prompts. The datasets are purely "extracted" from LLMs, with limited investigation into the appropriateness and quality of the LLM-generated instructions. Moreover, whether this approach is effective for non-English languages – often underrepresented in LLM training data – remains uncertain. We compare the effectiveness of LLM-generated instructions with human-written ones and propose a strategy for synthesizing instruction-tuning data for non-English languages.

## 7 Conclusion

Our work constructed datasets by leveraging open-weight LLMs to generate responses to human-written instructions. Three out of four LLMs fine-tuned with our datasets outperformed those fine-tuned with existing datasets built by self-synthesis, validating the usefulness of human-written instructions. Platforms like Chatbot Arena are thus essential for LLM development, serving not only as sites for manual evaluations but also as sources of real-world conversational data.

We applied the same strategy and synthesized state-of-the-art datasets for Japanese. With the datasets, we explore the cross-lingual transferability of instruction tuning. Abilities transferrable across languages turn out to be those requiring no factual knowledge, or global knowledge already present in the pre-trained models. Fundamental knowledge of a target language can only be enhanced by continuous pre-training.

While not explored in this study, we plan to enhance the utility of our datasets by incorporating support for safety and multi-turn conversations. Another future direction is to construct and evaluate the utility of synthesized datasets with instructions derived from human-written instructions that have been empirically validated as effective.

## Ethics Statement

This paper focuses more on improving the usefulness than the safety of LLMs. As a result, although we filtered out instructions flagged as toxic in LMSYS-Chat-1M, we acknowledge that **some toxic contents still remain in our datasets**. LLMs trained with our datasets should undergo further fine-tuning or reinforcement learning to ensure their safety and fairness. Derived from LMSYS-Chat-1M (Zheng et al., 2024), all instructions in our datasets have tokens that could reveal Personally Identifiable Information (PII) masked out. For LLM-generated responses, we leave them as-is, as the synthetic data generation process does not introduce any additional information about real individuals.

## Acknowledgments

This work was supported by JSPS KAKENHI Grant Number 25H01137. This work was also supported by JST K Program Japan Grant Number JPMJKP24C3. The research and development of the large language model Swallow has been supported by the AIST Project "Research and Development on Generative AI Foundation Models in the Physical Domain". It was also supported by a project from the Ministry of Education, Culture, Sports, Science, and Technology (MEXT) aimed at "establishment of research and development centers to ensure the transparency and reliability of generative AI models", along with other contributions. This research was conducted using the ABCI large-scale generative AI research and development support program, and the TSUBAME 4.0 supercomputer at Institute of Science Tokyo. We thank Sangwhan Moon for his valuable suggestions on improving the manuscript.

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

## A  License Information

Our datasets are granted for both research and commercial uses, in compliance with the LMSYS-Chat-1M Dataset License Agreement, together with the Llama 3.1 Community License Agreement[8] for Llama-based datasets and the Gemma Terms of Use[9] for Gemma-based datasets.

---

[8] https://www.llama.com/llama3_1/license/
[9] https://ai.google.dev/gemma/terms

| Base Model | SFT Dataset | MT-Bench | | |
| --- | --- | --- | --- | --- |
| | | Average | 1st Turn | 2nd Turn |
| *Reference: performance of teacher models* | | | | |
| Llama-3.1-405B-Instruct | | $8.33_{\pm.08}$ | $8.39_{\pm.08}$ | $8.27_{\pm.12}$ |
| Gemma-2-27B-IT | | $7.89_{\pm.05}$ | $8.19_{\pm.06}$ | $7.54_{\pm.10}$ |
| Qwen-2.5-7B | Magpie-Ultra-v0.1 | $3.56_{\pm.19}$ | $4.11_{\pm.22}$ | $2.96_{\pm.26}$ |
| | Magpie-Gemma2-Pro-Filtered | $6.00_{\pm.07}$ | $6.30_{\pm.08}$ | $5.66_{\pm.14}$ |
| | Magpie-Llama-3.1-Pro-MT-Filtered | $5.61_{\pm.05}$ | $6.33_{\pm.07}$ | $4.82_{\pm.07}$ |
| | Proposed-Llama-3.1-En | $\mathbf{7.10}_{\pm.12}$ | $\mathbf{7.67}_{\pm.06}$ | $\mathbf{6.47}_{\pm.21}$ |
| | Proposed-Gemma-2-En | $6.92_{\pm.06}$ | $7.59_{\pm.07}$ | $6.17_{\pm.13}$ |
| Gemma-2-9B | Magpie-Ultra-v0.1 | $5.99_{\pm.10}$ | $6.82_{\pm.09}$ | $5.09_{\pm.15}$ |
| | Magpie-Gemma2-Pro-Filtered | $5.81_{\pm.07}$ | $6.63_{\pm.05}$ | $4.83_{\pm.15}$ |
| | Magpie-Llama-3.1-Pro-MT-Filtered | $6.47_{\pm.08}$ | $7.23_{\pm.03}$ | $5.65_{\pm.16}$ |
| | Proposed-Llama-3.1-En | $\mathbf{6.79}_{\pm.06}$ | $\mathbf{7.39}_{\pm.07}$ | $\mathbf{6.13}_{\pm.06}$ |
| | Proposed-Gemma-2-En | $6.58_{\pm.06}$ | $7.11_{\pm.03}$ | $5.97_{\pm.12}$ |
| OLMo-2-1124-13B | Magpie-Ultra-v0.1 | $3.14_{\pm.05}$ | $4.42_{\pm.06}$ | $1.79_{\pm.08}$ |
| | Magpie-Gemma2-Pro-Filtered | $4.20_{\pm.15}$ | $5.66_{\pm.08}$ | $2.55_{\pm.27}$ |
| | Magpie-Llama-3.1-Pro-MT-Filtered | $\mathbf{6.85}_{\pm.10}$ | $7.16_{\pm.02}$ | $\mathbf{6.50}_{\pm.21}$ |
| | Tülu 3[†] | $5.92_{\pm.16}$ | $6.41_{\pm.13}$ | $5.38_{\pm.22}$ |
| | Proposed-Llama-3.1-En | $6.49_{\pm.08}$ | $\mathbf{7.50}_{\pm.04}$ | $5.39_{\pm.18}$ |
| | Proposed-Gemma-2-En | $5.44_{\pm.12}$ | $6.95_{\pm.11}$ | $3.75_{\pm.21}$ |

Table 5: Model performance evaluated on MT-Bench. † represents the official recipe utilized in the supervised fine-tuning of OLMo-2-1124-13B, with evaluation results of `allenai/OLMo-2-1124-13B-SFT` reported. This table is complementary to Table 2.

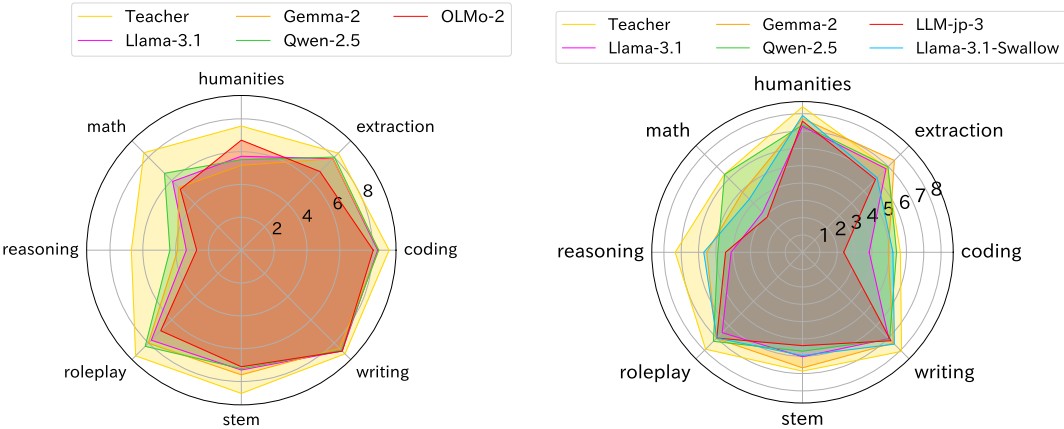

(a) Results on English MT-Bench. The teacher model is Llama-3.1-405B-Instruct.

(b) Results on Japanese MT-Bench. The teacher model is Gemma-2-27B-IT.

Figure 4: Radar chart of the performance of models for each task category in MT-Bench.

# B  Additional Experiments

This section presents additional experiments not included in Table 2, as shown in Table 5. Results for LMSYS-Chat-1M-Random, LMSYS-Chat-1M-Best (Zheng et al., 2024), and Ultra-Chat (Ding et al., 2023) are omitted, as they serve as a weak baseline intended for sanity checks; Models fine-tuned on these datasets significantly underperform those fine-tuned on Magpie (Xu et al., 2025) and our proposed datasets.

Table 5 shows trends consistent with Table 2, confirming the superiority of our proposed datasets over Magpie.

| SFT Data | # Instances | MT-Bench |
|---|---|---|
| **base model: Llama-3.1-8B** | | |
| Ja-Self-Instruct | 52,002 | 3.85 |
| Proposed-Gemma-2-Ja | 52,002 | **5.15** |
| **base model: llm-jp-3-13b** | | |
| Ja-Self-Instruct | 52,002 | 4.91 |
| Proposed-Gemma-2-Ja | 52,002 | **5.64** |
| **base model: Llama-3.1-Swallow-8B-v0.1** | | |
| Ja-Self-Instruct | 52,002 | 4.53 |
| Proposed-Gemma-2-Ja | 52,002 | **5.71** |

Table 6: Performance of LLMs evaluated on MT-Bench when the number of training instances is aligned. Fine-tuning on our dataset, even downsampled, outperforms those fine-tuned on Ja-Self-Instruct.

## C Category-Wise Effectiveness

MT-Bench includes instructions across eight categories. Here, we examine how our collected datasets enhance performance in each category. Evaluation results for LLMs tuned on Proposed-Llama-3.1-En and Proposed-Gemma-2-Ja are shown in Figure 4a and 4b, respectively. The performance of teacher models, i.e., LLMs utilized for generating responses, is also included to see how the student models inherited the capabilities.

Firstly, we observe a similar tendency in both radar charts that our datasets effectively enhance LLMs' abilities in writing, coding, and extraction. Meanwhile, a common difficulty has been observed in inheriting the ability to reason. The results indicate that **imitating the response generated from stronger LLMs can be insufficient to equip weaker models with reasoning abilities**.

On the other hand, in the math category for both languages, we observe that Qwen-2.5 leads in performance. The base models in the Qwen series are known for their strength in math tasks. Similarly, in the Japanese humanities category (Figure 4b), Llama-3.1-Swallow outperforms Llama-3.1 and demonstrates the best performance among all student models. Llama-3.1-Swallow, which was continuously pre-trained with Japanese corpus on Llama-3.1, gains a deeper understanding of Japanese local culture than Llama-3.1 (Okazaki et al., 2024; Fujii et al., 2024). These results suggest that **the inherent advantages in reasoning abilities and knowledge from base models persist even after instruction-tuning**.

Based on the above observations, we conclude that instruction-tuning by imitating the responses of strong LLMs is effective for tasks related to writing or reformatting. However, its effectiveness is limited for tasks that require deep knowledge and reasoning. In these cases, the advantages of the base models remain even after instruction-tuning.

## D Instance-Wise Effectiveness: Japanese

**Settings.** To match the dataset scale of Ja-Self-Instruct, we randomly sampled 52K (*instruction*, *response*) pairs from Proposed-Gemma-2-Ja. LLMs were fine-tuned on this subset and compared with those fine-tuned on Ja-Self-Instruct. The results are shown in Table 6.

**Results.** LLMs fine-tuned on our dataset outperform those fine-tuned on Ja-Self-Instruct, even when the number of training instances is aligned to the same. Compared to English settings, the performance gap between our datasets and the existing ones is more pronounced in the Japanese settings, underscoring a significant shortage of Japanese resources for instruction-tuning.

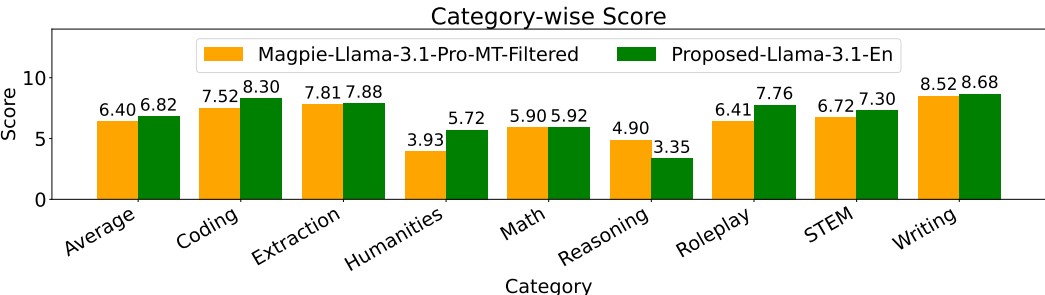

Figure 5: Category-wise scores of models fine-tuned on Magpie-Llama-3.1-Pro-MT-Filtered and Proposed-Llama-3.1-En. The performance gap is larger in categories underrepresented in Magpie's topic distribution.

# E   Category-Wise Quantitive Analysis

Corresponding to Section 5.1, here we report the category-wise performance of Llama-3.1-8B trained on Magpie-Llama-3.1-Pro-MT-Filered and Proposed-Llama-3.1-En in Figure 5.

As shown in the plot, models trained on our dataset outperform those trained on Magpie in the *Humanities* and *Roleplay* categories, while the advantage diminishes in *Reasoning* and *Math*. This trend may reflect the topic distributions discussed in Section 5.1, where our dataset is more balanced but Magpie is heavily skewed toward mathematics. The skew likely boosts performance in Reasoning and Math, consistent with recent findings on the close relationship between mathematical and reasoning abilities (DeepSeek-AI, 2025).

