# OpenReview forum: "Building Instruction-Tuning Datasets from Human-Written Instructions with Open-Weight Large Language Models"
_colmweb.org/COLM/2025/Conference — COLM 2025_

### Official Review · Reviewer_UvMA · 2025-04-22

**Rating:** 6
**Confidence:** 5
**Ethics Flag:** 1

**Summary:**

This paper investigates whether human-written instructions are still necessary for instruction tuning of large language models (LLMs), despite recent success of fully synthetic datasets like Magpie. The authors build new instruction-tuning datasets by pairing real user-written instructions from LMSYS-Chat-1M with responses generated by state-of-the-art open-weight LLMs (Llama-3.1 and Gemma-2). Fine-tuning on these datasets consistently outperforms existing instruction-tuning datasets—including Magpie—on MT-Bench evaluations across multiple base models. The authors further demonstrate that their approach generalizes to other languages, such as Japanese, by translating instructions and generating corresponding responses. They show that instruction-tuning improves instruction-following ability, while culturally specific knowledge still requires language-specific pretraining. The paper concludes that combining human-written instructions with LLM-generated responses yields superior datasets and presents a scalable and open-source method for instruction-tuning across languages.

**Reasons To Accept:**

1. The paper demonstrates that datasets built by pairing human-written instructions with responses from open-weight LLMs consistently outperform existing state-of-the-art datasets (e.g., Magpie) across multiple base models (Llama, Gemma, Qwen, OLMo) and in multiple languages (English and Japanese). This highlights the robustness and generalizability of their approach.

2. The authors present a simple yet effective and reproducible pipeline for instruction tuning: using real-world human-chatbot conversations (LMSYS-Chat-1M) for instructions and generating responses using publicly available LLMs. Their datasets are openly licensed, allowing for broader adoption in both academic and commercial settings.

3. The paper goes beyond just performance metrics and provides thoughtful analysis of cross-lingual instruction tuning. It distinguishes between capabilities gained through instruction tuning and those requiring continuous pretraining, offering valuable insights into the limitations of current models in handling culture-specific knowledge.

**Reasons To Reject:**

1. The primary evaluation relies heavily on MT-Bench, which, while strong, may not comprehensively capture all facets of instruction-following (e.g., safety, robustness, or factual accuracy). The paper could be strengthened by including human evaluations or a broader range of benchmarks to support its claims more holistically.

2. The proposed datasets are primarily single-turn, while many real-world applications of LLMs involve multi-turn interactions. Although the paper acknowledges this and cites it as future work, the lack of multi-turn data may limit the practical applicability of the datasets in dialog-heavy tasks.

3. The cross-lingual extension to Japanese relies on machine-translating English instructions. This approach may introduce translation artifacts or overlook linguistic and cultural nuances, potentially affecting instruction quality. A more thorough analysis of translation quality or a comparison with native Japanese instructions would strengthen the multilingual claims.

---

> ### Author Response · Authors · 2025-05-30
>
> Thank you for your time and effort in reviewing our manuscript! Your positive comments mean a great deal to us. We respond to your comments as follows, in the hope of addressing your concerns.
>
> > The primary evaluation relies heavily on MT-Bench, which, while strong, may not comprehensively capture all facets of instruction-following (e.g., safety, robustness, or factual accuracy). The paper could be strengthened by including human evaluations or a broader range of benchmarks to support its claims more holistically.
>
> We would like to clarify that this work focuses on the "helpfulness" of LLMs, i.e., how they provide helpful and informative responses to user requests. Other facets, specifically "safety" or "harmlessness", have been recognized as having a trade-off with helpfulness [1]. This study investigates **how human-originated signals contribute to stimulating LLMs to provide informative responses, without interference from refusing-to-answer training related to safety guards**. As mentioned in L.326, we plan to consider safety as the next step.
>
> Regarding the evaluation of helpfulness, as mentioned in L.115 - 117, assessments from LLMs and humans on MT-Bench have been validated to highly agree with each other. Specifically, in [2], the authors reported that "strong LLM judges like GPT-4 can match both controlled and crowdsourced human preferences well, achieving over 80% agreement, the same level of agreement between humans". Other benchmarks, such as AlpacaEval 2.0, also evaluate the helpfulness of LLMs, but MT-Bench correlates to human preferences as well as these benchmarks [3]. Our evaluation setting thus arguably provides a comprehensive assessment of LLMs. We will make this clearer in the final manuscript.
>
> > The proposed datasets are primarily single-turn, while many real-world applications of LLMs involve multi-turn interactions. Although the paper acknowledges this and cites it as future work, the lack of multi-turn data may limit the practical applicability of the datasets in dialog-heavy tasks.
>
> The ability to engage in multi-turn conversation is indeed an important ability of LLMs, and as mentioned in the manuscript, we plan to extend our datasets to multi-turn in future work.
>
> At the same time, we would like to highlight that three out of four models trained on our dataset exhibit higher second-turn performance compared with those trained on MAGPIE, which is a multi-turn dataset (Table 2). These results demonstrate that **models are capable of acquiring multi-turn conversational abilities through training on our datasets, even though they are single-turn**. This highlights the practical applicability of our dataset even in multi-turn scenarios.
>
> > The cross-lingual extension to Japanese relies on machine-translating English instructions. This approach may introduce translation artifacts or overlook linguistic and cultural nuances, potentially affecting instruction quality. A more thorough analysis of translation quality or a comparison with native Japanese instructions would strengthen the multilingual claims.
>
> We would like to respectfully mention that **we have compared our translated dataset with a competitor sourced from native-quality instructions**. The competitor is Ja-Self-Instruct in Tables 4 and 5, an instruction-tuning dataset synthesized from Japanese seed instructions post-edited to meet native quality [4]. As shown in these tables, models trained on our datasets outperform those trained on Ja-Self-Instruct.
>
> Nevertheless, we would also like to emphasize that **obtaining native instructions is challenging**. While native instructions would certainly be ideal, there is no clear best practice for determining which types of instructions are necessary, making the collection of manual instructions from scratch difficult. With our method, instruction-tuning datasets of reasonable quality can be built without incurring additional costs for inviting human annotators or designing the instruction scheme. The quality of instructions could indeed be affected by the translator, but **we have demonstrated that the instruction-translated, response-generated approach can yield high-quality datasets (even better than the self-instruct approach based on native-quality seed instructions) for non-English languages**. We will make this claim clearer in the final manuscript.
>
> [1] Llama 2: Open Foundation and Fine-Tuned Chat Models. Touvron et al. arXiv:2307.09288.
> [2] Judging LLM-as-a-Judge with MT-Bench and Chatbot Arena. Zheng et al. NeurIPS 2023.
> [3] Length-Controlled AlpacaEval: A Simple Way to Debias Automatic Evaluators. Dubois et al. COLM 2024.
> [4] Rapidly Developing High-quality Instruction Data and Evaluation Benchmark for Large Language Models with Minimal Human Effort: A Case Study on Japanese. Sun et al. LREC-COLING 2024.

---

> ### Author Response · Authors · 2025-06-03
> **We Are Open to Discussions**
>
> Dear Reviewer UvMA,
>
> Again, thank you for taking the time to review our paper and for your positive and thoughtful feedback. As the final stage of the discussion period has begun, we would like to express our sincere interest in discussing with you. Below we have provided our responses, in the hope of addressing your concerns. We would greatly appreciate it if you could review the responses and let us know if they are clear enough. Also, please feel free to share any further concerns you may have. We look forward to your reply.

---

### Official Review · Reviewer_DRap · 2025-05-12

**Rating:** 7
**Confidence:** 3
**Ethics Flag:** 1

**Summary:**

This work contribution is two-fold: 1) it studies the performance of instruction-tunining of open-source LLMs on synthetically generated data by large open-source teacher model. It gets inspiration from Magpie [1] work that demonstrates that it is possible to extract instructions and responses from pretrained open-source LLMs by prompting with  basic templates.  This work suggest to use human-generated instructions, and extract only responses from pretrained open-source LLMs. Authors demonstrate that relying on human generated instructions lead to superior performance comapred to Magpie approach, and can also sometimes lead to better performance that models tuned on human-generated responses.  Second part of this work extends such synthetic data generation to Japanese language and demonstrates that it is possible to build on strong teacher models distillation to create competitive japanes instruciton-tuned models.  It also perform some interesting analysis on the importance of continous pretraining and instruction finetuning in non-english language (Japanese in this case).


[1] Magpie: Alignment Data Synthesis from Scratch by Prompting Aligned LLMs with Nothing Xu et al. ICLR 2025

**Questions To Authors:**

- Did you assess how good gpt-4 is at assessing answers in japanese?
- You mention to skip some of the results due to page limit. I believe good practice is to put those results in the appendix for the interested reader. It allows to get more complete picture of the work.
- discussion about discrepancy in student ant teacher models behaviour for English and Japanes (lines 232-234): did you manually examine the performance of both teacher models? Did you observe presence of code-switching? How does gpt4-based judge behave on code-switching data? In our previous experiments we observed that and answer  would be judged as correct by gpt4o, even if it is not generated in the same language as question. If one of the teachers tends to generate more code-switched answers than other it won't necessarily be reflected in overal score computed by gpt4 judge, but can have an important impact on the performance of instruction tuned model on such dataset.

**Reasons To Accept:**

- This work reports results outperforming Magpie work, which might be of interest for future LLMs improvements with synthetic datasets
- Results on Japanses instruction finetuning are interesting, as well as analysis on the continout pretraining + instruction tuning across different domains in japanese.

---
Score updated after clarifications provided by the authors

**Reasons To Reject:**

- It seems to me that the findings of the first part of this  work are quite marginal compared to previous studies in synthetic data generation [1].  More in-depth discussion and analysis of the results comparing Magpie to the proposed method would help to strengthen this contribution: eg perform some manual analysis of the examples where proposed method outperforms magpie and vice versa would be interesting. It could also be possible to trace back the predicitons to the training data similar to OlmoTrace work. That would allow to perform even more rich analysis.

- The finetuning on Japanese synthetic data and analysis is interesting, but clearly lacks more in depth analysis and comparison to previous studies in multilingual instruction tuning [3,4,5]. Assessment of reliability of LLM-as  a judge for Japanese seems important as well: eg. how significant would be the difference of 0.2-1 in scores if we take llms-as-a-judge potential error into account?


[1] Magpie: Alignment Data Synthesis from Scratch by Prompting Aligned LLMs with Nothing Xu et al. ICLR 2025
[2] Multilingual Instruction Tuning With Just a Pinch of Multilinguality Shaham et al. Findings of ACL 2024
[3] Zero-shot cross-lingual transfer in Instruction Tuning of large language models. Chirkova et al. INLG 2024
[4] Turning English-centric LLMs Into Polyglots: How Much Multilinguality Is Needed? Kew et al. EMNLP 2024

---

> ### Author Response · Authors · 2025-05-30
> **Response 1/3: Contributions and Discussions of our Datasets v.s. Magpie**
>
> Thank you for your time and effort in reviewing our manuscript! We are truly encouraged by your recognition of our efforts in Japanese instruction tuning. Below, we respond to your comments in the hope of addressing your concerns.
>
> > It seems to me that the findings of the first part of this work are quite marginal compared to previous studies in synthetic data generation [1]. More in-depth discussion and analysis of the results comparing Magpie to the proposed method would help to strengthen this contribution (...)
>
> An important contribution of the first part of our work is **demonstrating that human-originated signals are still important for effective instruction-tuning**. Magpie synthesizes datasets from "nothing", generating instructions solely from the instruction prefix. We are interested in whether this pure-synthetic approach can outperform the mainstream approaches based on human-originated signals, as claimed in the paper [1]. Through experiments, we find out that human-written instructions in LMSYS-Chat-1M are superior to those synthesized in Magpie, verifying the importance of human-originated signals. We believe **these findings can offer valuable insights for future LLM R&D when building instruction-tuning datasets**. We will make this idea clearer in the final version of our manuscript.
>
> For more in-depth analyses of our datasets and Magpie, we present two quantitative analyses here, which can also be included in the final manuscript:
>
> **(1) Top-10 most frequent topics of instructions**
>
> We assign tag(s) to each instruction using InsTag [3], a tool designed to analyze the semantics and intentions of instructions. These tags help quantify the diversity of the instruction set. The top 10 most frequent tags and their corresponding frequencies are shown below.
>
> | # Magpie-Llama-3.1-Pro-MT-Filtered |  |  | # Proposed-Llama-3.1-En |  |
> | --- | --- | --- | --- | --- |
> | Instruction tag | frequency |  | Instruction tag | frequency |
> | mathematics | 1,520 |  | code example | 193 |
> | problem solve | 530 |  | information request | 185 |
> | geometry | 463 |  | information retrieval | 176 |
> | physic | 255 |  | company introduction | 168 |
> | algebra | 243 |  | python programming | 166 |
> | information request | 192 |  | article write | 146 |
> | measurement | 184 |  | spelling and grammar check | 128 |
> | data analysis | 150 |  | location | 125 |
> | word problem | 133 |  | mathematics | 115 |
>
> From the table, we observe that Magpie synthetic instructions are skewed toward mathematics and related topics, while our dataset has a more balanced topic distribution across various categories, including article writing, grammar check, etc. This broad coverage could be a potential reason why our dataset more effectively enhances the general instruction-following ability of LLMs.
>
> **(2) Category-wise performance on the MT-Bench of trained models**
>
> MT-Bench covers 8 different categories: Coding, Extraction, Humanities, Math, Reasoning, Roleplay, STEM, and Writing [4]. Here we report the category-wise performance of Llama-3.1-8B trained on Magpie and trained on our dataset.
>
> | dataset                              | Average | Coding | Extraction | Humanities | Math | Reasoning | Roleplay | STEM | Writing |
> |--------------------------------------|---------|--------|------------|------------|------|-----------|----------|------|---------|
> | (1) Proposed-Llama-3.1-En                | **6.82**    | **8.30**   | **7.88**       | **5.72**       | **5.92** | 3.35      | **7.76**     | **7.30** | **8.68**    |
> | (2) Magpie-Llama-3.1-Pro-MT-Filtered     | 6.40    | 7.52   | 7.81       | 3.93       | 5.90 | **4.90**     | 6.41     | 6.72 | 8.52    |
> | (1) - (2)                            | 0.42 | 0.78 | 0.07 | 1.79 | 0.02 | -1.55 | 1.35 | 0.58 | 0.16 |
>
> From the table, compared to training on Magpie, training on our dataset shows clear superiority in Humanities and Roleplay, while the superiority disappears in Reasoning and Math. This trend is potentially related to the topic distribution discussed above, where our proposed dataset has a more balanced topic distribution, while Magpie is skewed towards mathematics. The skew could make the model perform better in Reasoning and Math compared to other categories, in line with recent findings that reasoning and math abilities are closely related [5].
>
> [1] Magpie: Alignment Data Synthesis from Scratch by Prompting Aligned LLMs with Nothing. Xu et al. ICLR 2025.
> [2] Self-Instruct: Aligning Language Models with Self-Generated Instructions. Wang et al. ACL 2023.
> [3] #InsTag: Instruction Tagging for Analyzing Supervised Fine-tuning of Large Language Models. Lu et al. ICLR 2024.
> [4] Judging LLM-as-a-Judge with MT-Bench and Chatbot Arena. Zheng et al. NeurIPS 2023.
> [5] DeepSeek-R1: Incentivizing Reasoning Capability in LLMs via
> Reinforcement Learning. DeepSeek-AI.

---

> > ### Author Response · Authors · 2025-05-30
> > **Response 2/3: Existing Multilingual Instruction Tuning Approaches Suffer from Off-Target Issue, Usefulness of GPT-4-as-a-judge in Japanese MT-Bench**
> >
> > > The finetuning on Japanese synthetic data and analysis is interesting, but clearly lacks more in depth analysis and comparison to previous studies in multilingual instruction tuning [3,4,5].
> >
> > Thank you for highlighting the related studies on multilingualism. However, we have found that **existing multilingual instruction tuning approaches tend to undermine end-user usefulness due to the off-target issue** [1]—namely, the unnatural switching from the target language (Japanese, in our case) to a non-target language such as English. Typically, multilingual instruction tuning involves mixing a relatively small portion of non-English data with predominantly English data. In our experiments, this led to undesirable code-switching when training on a mixture of English and Japanese data. For instance, in the example below, the response abruptly switches from Japanese to English (`This headline highlights ...`).
> > ```
> > Instruction：最新の日本の太陽光発電技術に関する記事のための、科学的に正確かつ興味を引く見出しを4つ提案してください。​
> > Output：日本の太陽光発電技術に関する、科学的に正確で興味を引く見出しを4つご紹介します。​
> > 日本の太陽光発電革命：次世代パネルの登場で再生可能エネルギーの未来が変わる！ (This headline highlights the cutting-edge developments in solar panel technology and their impact on Japan's renewable energy landscape.)
> > ```
> >
> > One of our intentions is to help **develop LLMs with strong proficiency in a local language that are useful for end-users**. We are therefore highly sensitive to code-switching and thus did not adopt multilingual approaches. We will include these discussions and clarify our focus on language adaptation in the final manuscript.
> >
> > > Assessment of reliability of LLM-as a judge for Japanese seems important as well: eg. how significant would be the difference of 0.2-1 in scores if we take llms-as-a-judge potential error into account?
> >
> > We provide evidence for the statistical significance of GPT-4-as-a-judge in response to your first question in the Section Questions to Authors. Below, we report the standard deviation of evaluation results on Japanese MT-Bench across five runs, as mentioned in Lines 120–121. Under this setting, the standard deviation is approximately 0.15, providing evidence that **a difference greater than 0.3 on the Japanese MT-Bench can be considered statistically significant**.
> >
> > | model   | IT dataset | avg score of 5 runs |std dev of 5 runs|
> > |---------|---------|---------|---------|
> > | Gemma-2-27b-IT                | --|7.10    |   0.06 |
> > | Llama-3.1-405B-Instruct                  | --|6.79    | 0.17 |
> > | Llama-3.1-8B   | Proposed-Llama-3.1-Ja|5.05    |  0.16  |
> > | Llama-3.1-8B   | Proposed-Gemma-2-Ja | 5.64  | 0.07   |
> > |  Gemma-2-9B    | Proposed-Llama-3.1-Ja |5.14   |  0.16  |
> > |  Gemma-2-9B  |  Proposed-Gemma-2-Ja |6.49 |  0.17   |
> > | Llama-3.1-Swallow-8B-v0.1 | Proposed-Llama-3.1-Ja  |  5.18 | 0.05  |
> > | Llama-3.1-Swallow-8B-v0.1 |Proposed-Gemma-2-Ja  | 6.23 | 0.12  |
> >
> > > Did you assess how good gpt-4 is at assessing answers in japanese?
> >
> > Yes, we have evidence that GPT-4 can successfully assess answers in Japanese.
> >
> > **Evidence 1: Chatbot Arena LLM Leaderboard**
> >
> > The Japanese MT-Bench rankings (shown in Table 4) are also consistent with the **Japanese category** of the Chatbot Arena Leaderboard [2] as of 2025-05-30. Chatbot Arena is "an open-source platform for evaluating AI through human preference, developed by researchers at LMArena. With over 1,000,000 user votes, the platform ranks the best LLM and AI chatbots using the Bradley-Terry model to generate live leaderboards" (cited from the page).
> > Specifically, Gemma-2-27B-IT is ranked #32, Llama-3.1-405B-Instruct-fp8 is #33, Gemma-2-9B is #46, and Llama-3.1-8B is #71.
> >
> > **Evidence 2: Chatbot Arena hosted locally in Japan**
> >
> > Assessments of GPT-4 on Japanese MT-Bench have been shown to have a strong correlation (pearson correlation = 0.642) to human assessments (Elo rating) in a recent experiment called LLM-jp Chatbot Arena [3]. In the page, the organizers noted that the Elo rating was not converged yet due to a small sample size (n=1,003), and that the correlation may increase once the Elo rating converges.
> >
> > Therefore, we believe that using GPT-4-as-a-judge is a valid choice in assessing answers in Japanese.
> >
> > > You mention to skip some of the results due to page limit. I believe good practice is to put those results in the appendix for the interested reader. It allows to get more complete picture of the work.
> >
> > Thank you for the valuable suggestion. We will include the experiment results omitted in Table 2 in the Appendix of the final manuscript.
> >
> > [1] Sennrich et al. Mitigating Hallucinations and Off-target Machine Translation with Source-Contrastive and Language-Contrastive Decoding. In: EACL 2024.
> > [2] https://huggingface.co/spaces/lmarena-ai/chatbot-arena-leaderboard, please change the `Category` tag to `Japanese`
> > [3]  https://llm-jp.nii.ac.jp/ja/blog/blog-836/, the page is in Japanese

---

> > > ### Author Response · Authors · 2025-05-30
> > > **Response 3/3: We have paid great attention to code-switching issues for both training and evaluation stages**
> > >
> > > > discussion about discrepancy in student ant teacher models behaviour for English and Japanes (...)
> > >
> > > Thank you for highlighting the issue of code-switching. We are well aware of this potential issue and have paid careful attention to avoid introducing code-switched texts into our dataset. Details below.
> > >
> > > > did you manually examine the performance of both teacher models? Did you observe presence of code-switching?
> > >
> > > Yes, we have manually reviewed the responses generated by both teacher models. We have 2 native Japanese speakers (or speakers at a native level) with PhD degrees in NLP to evaluate the Japanese responses, where **almost no code-switching (fewer than 0.5% of the instances)** was observed for both teacher models.
> > >
> > > > How does gpt4-based judge behave on code-switching data?
> > >
> > > For the Japanese MT-Bench, we adopt the judging prompt used in Nejumi leaderboard​ Neo [1], a leaderboard organized by Weights & Biases Japan to evaluate Japanese LLMs. The prompt includes language-specific instructions as below. We have verified that it appropriately penalizes responses that are not in Japanese.
> > >
> > >
> > > ```
> > > # Excerpt from judging prompt
> > > … The expected language is Japanese. Responses in languages other than Japanese will incur score deductions unless specifically required. Failure to use Japanese at all will result in the lowest evaluation. However, using Japanese is not mandatory when providing only Python scripts or calculation results, where Japanese is not essential. Additionally, your explanation of judgement should be in Japanese.
> > > ```
> > > Additionally, we carefully monitor whether our trained models produce code-switched texts by measuring the Japanese-character ratio of generated outputs. We attach the Japanese-character ratio of the trained models below. (Note that the Japanese character ratio is lower than one might intuitively expect, as code snippets are counted as non-Japanese characters.)
> > >
> > > | reference  | JMT Avg | Ja char ratio |
> > > |---------|---------|---------|
> > > | Llama-3.1-Instruct-405B-Instruct | 6.79   |  67.1%  |
> > > | gemma-2-27b-it | 7.10   |  66.2%  |
> > >
> > > Base model: Llama-3.1-Swallow-8B-v0.1
> > > | teacher model  | JMT Avg | Ja char ratio |
> > > |---------|---------|---------|
> > > | Llama-3.1-Instruct-405B-Instruct     |   5.18  |  64.9%  |
> > > | gemma-2-27b-it |  6.23  |   69.3% |
> > >
> > >
> > > From the table, we can see that using llama-3.1 and gemma-2 as the teacher model yields models with similar Japanese character ratios. The ratio is also close to that for Llama-3.1-Instruct-405B-Instruct and gemma-2-27b-it. We will include the results in the final version of the manuscript.
> > >
> > > [1] https://wandb.ai/wandb-japan/llm-leaderboard/reports/Nejumi-LLM-Neo--Vmlldzo2MTkyMTU0

---

> ### Author Response · Authors · 2025-06-03
> **Thank you for acknowledging our response!**
>
> Dear Reviewer DRap,
>
> Thank you for acknowledging our response and reconsidering the score! Your positive feedback truly means a lot to us. We will be sure to incorporate the discussions mentioned in the rebuttal into the final manuscript.

---

> ### Comment · Reviewer_DRap · 2025-06-09
> **Thank you for your answers**
>
> Dear authors,
>
> thank you for your answers and for providing additional details.
>
> Based on your answers I've updated the score since they made it all more clear and more convicing. I believe that it would be really beneficial for the paper if you add some of these results in the main paper, and some of the discussions/results in the appendix to make it more self-contained and to improve the readability.

---

> > ### Author Response · Authors · 2025-06-10
> > **Thank you for the reply!**
> >
> > Dear reviewer DRap,
> >
> > Thank you for responding to us and reconsidering the score! Based on your valuable suggestions, we will definitely include the discussions in the final version of the manuscript to make things clearer.
> >
> > Again, we appreciate your time and patience spent on our manuscript.

---

### Official Review · Reviewer_WhHW · 2025-05-13

**Rating:** 6
**Confidence:** 3
**Ethics Flag:** 1

**Summary:**

The main research question that is explored in the paper is whether "LLM-generated instructions are better than human-written ones" . The paper presents a dataset that is created using human-written instructions from an instruction dataset coupled with generated responses from open LLMs. The results of fine-tuning are compared to existing synthetic datasets such as Magpie (Xu et al) among other baselines. The paper shows that instruction tuning with the proposed dataset achieves superior or comparable performance to the baselines and prior work. A controlled experiment in terms of the dataset size by downsampling the proposed dataset to the same size of Magpie is also conducted and shows superior results. It is observed that finetuning with the proposed dataset also has gains to Japanese.

Furthermore, machine translation is employed to translate the instruction datasets to Japanese to evaluate performance of the method is Japanese.

However, the key research question remains largely unanswered. A more controlled experiment with the human-written instructions of the same dataset substituted with its synthetic counter part which could be generated using the procedure adopted in Magpie would have been a more appropriate baseline.

**Questions To Authors:**

* Adding more languages beyond Japanese would  enable authors to make more generalized conclusions regarding finetuning on non-English languages.

**Reasons To Accept:**

* The research question is interesting. It is valuable to the community to know the tradeoffs of various approaches to generate synthetic instruction tuning datasets that are critical for aligning language models.

* The exploration of an additional language beyond English in the experiments is a value add.

**Reasons To Reject:**

* The main research question remains largely unanswered since the baselines are derived from other data sources. A more controlled experiments using the same seed LM-Sys-Chat dataset with synthesised instructions is needed to answer the research question of whether human-written instructions are more valuable that machine generated instructions.

---

> ### Author Response · Authors · 2025-05-30
>
> Thank you for your time and effort in reviewing our manuscript! However, we are concerned that **there might be some misunderstanding regarding our motivation and research question**. Below, we respond to your comments in the hope of clearing up any misunderstandings and addressing your concerns.
>
> > The main research question remains largely unanswered since the baselines are derived from other data sources. A more controlled experiments using the same seed LM-Sys-Chat dataset with synthesised instructions is needed to answer the research question of whether human-written instructions are more valuable that machine generated instructions.
> >However, the key research question remains largely unanswered. A more controlled experiment with the human-written instructions of the same dataset substituted with its synthetic counter part which could be generated using the procedure adopted in Magpie would have been a more appropriate baseline.
>
> We would like to clarify that our research question, as consistently stated in L.4-5 and L.32, is to investigate whether "human-originated signals" are necessary for building instruction-tuning datasets. This research question arises from Magpie, a method that generates instructions solely from the instruction prefix (i.e., `<|start_header_id|>\nuser\n <|end_header_id|>`). We are curious about **whether the pure-synthetic approach truly outperforms mainstream methods based on human-originated signals, as claimed in the Magpie paper** [1]. With this question in mind, we conduct experiments that compare Magpie with our LMSYS-Chat-1M dataset, comprising human-written instructions and LLM-generated responses. The results highlight the effectiveness of human-written instructions, providing a clear answer to our research question: human-originated signals are still necessary. We will clarify this research question more explicitly in the final manuscript.
>
> Furthermore, we would like to politely point out that the approach you mentioned in the summary – `A more controlled experiment with the human-written instructions of the same dataset substituted with its synthetic counter part which could be generated using the procedure adopted in Magpie would have been a more appropriate baseline` – is infeasible: Magpie generates instructions from the instruction prefix and does not support synthesizing based on given instructions by design.
> Admittedly, it is worth exploring whether the effectiveness of human-originated signals can be enhanced using LLM-driven instruction synthesis methods such as Self-Instruct [2] or Eval Instruct [3] on LMSYS-Chat-1M human instructions as seeds. However, **this lies beyond the scope of our research question**, especially given that “its (synthesizing from seed instructions) success heavily depends on prompt engineering and the careful selection of initial seed questions” [1]. We believe that **using human-written instructions as-is, as we have done in this research, is a straightforward and sufficient approach to answer our research question**.
>
> > Adding more languages beyond Japanese would enable authors to make more generalized conclusions regarding finetuning on non-English languages.
>
> Thank you for your valuable suggestions. While deriving best practices for finetuning on non-English languages is not the main focus of this study, we agree that conducting experiments in additional languages would enhance the robustness of our findings. It is potentially interesting to extend our approach to more non-English languages.
>
> [1] Xu et al. Magpie: Alignment data synthesis from scratch by prompting
> 476 aligned llms with nothing. In ICLR 2025.
> [2] Wang et al. Self-Instruct: Aligning Language Models with Self-Generated Instructions. ACL 2023.
> [3] Xu et al. WizardLM: Empowering large pre-trained language models to follow complex instructions. In ICLR 2024.

---

> > ### Comment · Reviewer_WhHW · 2025-06-09
> >
> > > whether the pure-synthetic approach truly outperforms mainstream methods based on human-originated signals, as claimed in the Magpie paper [1]
> >
> > Thank you for response. I think the motivation and research question was very clear from the paper and it was well understood. The question is, "whether "human-originated signals" are necessary for building instruction-tuning datasets."  To this end, human-written instructions were paired with LLM-based responses from LM-Sys-Chat-1M dataset.
> >
> > The concern that was raised in the review was related to using the a baseline with the same dataset LM-Sys-Chat-1M with the LLM responses, but the instructions regenerated synthetically (possibly given the response alone). This would be useful to ascertain whether human-written instructions are necessary in instruction-tuning datasets. Although I agree that such a dataset has already used human-originated "signals" to generate LLM responses in the first place.  Comparing to other methods on similar-sized dataset alone (but from a different data source) does not conclude as the domain or topics in the data may also have an impact on the results (Table 3). Nonetheless I understand the difficulty in running such a controlled experiment, and will raise my scores accordingly.

---

> > > ### Author Response · Authors · 2025-06-09
> > > **Thank you for acknowledging our response!**
> > >
> > > Thank you for acknowledging our response and reconsidering the score! Your feedback truly means a lot to us. We are relieved to know that our claim in the paper is clear, and we appreciate the clarification regarding the concern raised in the review.
> > >
> > > > The concern that was raised in the review was related to using the a baseline with the same dataset LM-Sys-Chat-1M with the LLM responses, but the instructions regenerated synthetically (possibly given the response alone). This would be useful to ascertain whether human-written instructions are necessary in instruction-tuning datasets.
> > >
> > > The experiments you mentioned are very interesting to us. However, as noted in your response, these experiments would still involve "human-originated signals" at the outset, which fall outside the scope of our current work. Nevertheless, we agree that such experiments would be valuable in distinguishing the effect of the instructions' surface form from their semantics (topics). We are definitely interested in pursuing these analyses in the future.
> > >
> > > Again, thank you for your time and patience, not only in reviewing our manuscript but also in engaging in this discussion.

---

> ### Author Response · Authors · 2025-06-03
> **We Are Open to Discussions**
>
> Dear Reviewer WhHW,
>
> Again, thank you for taking the time to review our paper. As the final stage of the discussion period has begun, we would like to express our sincere interest in discussing with you. Below we have provided our responses, in the hope of addressing your concerns. We would greatly appreciate it if you could review the responses and let us know if they are clear enough. Also, please feel free to share any further concerns you may have. We look forward to your reply.

---

> ### Author Response · Authors · 2025-06-08
> **Gentle Reminder: The discussion period will end shortly, and we are awaiting for your reply**
>
> Dear Reviewer WhHW,
>
> As the discussion period will end in 72 hours, we would like to politely send a reminder and express our hope to hear back from you.
>
> We are sending this special reminder to you because, based on your comments, we sense there might be some misunderstandings regarding the motivation and research question of our work. We are concerned that these misunderstandings might have impacted the evaluation of our paper’s contribution, as well as the rating.
>
> To clear up these potential misunderstandings, we have provided clarifications in the response below. We would greatly appreciate it if you could review the response and let us know if it is clear. If you have any further questions, we are more than happy to continue the discussion with you.
>
>
> Again, thank you in advance for your time and patience.
> We are looking forward to hearing back from you.

---

### Official Review · Reviewer_7i5D · 2025-05-23

**Rating:** 7
**Confidence:** 4
**Ethics Flag:** 1

**Summary:**

The authors introduce a method to generate instruction-tuning datasets using human
instructions from a real-world dataset. The method is tested in several LLMs and performs
well. The method is evaluated in English and Japanese.

**Questions To Authors:**

Line 106 – Should this say “maximize log probability” instead of minimize?
Line 212 – Should be “one of few”
Line 315 – “Four” should not be capitalized

**Reasons To Accept:**

The paper is very well written and is easy to follow. The authors test on a variety of
LLMs and test in English and Japanese. The method compares favorably with Magpie,
a SOTA instruction-tuning dataset.

**Reasons To Reject:**

A relatively minor weakness is that only one alternative instruction-tuning dataset,
Magpie, is examined.

---

> ### Author Response · Authors · 2025-05-30
>
> Thank you for your time and effort in reviewing our manuscript! Your positive comments mean a great deal to us and have truly encouraged us. We will incorporate your suggestions into the final manuscript. We respond to your comments as follows, in the hope of addressing your concerns.
>
> > A relatively minor weakness is that only one alternative instruction-tuning dataset, Magpie, is examined.
>
> We chose Magpie as our main competitor for two reasons.
>
> Firstly, **Magpie is closely aligned with our research question**. It is an instruction-tuning dataset synthesized from "nothing", generating instructions solely from the instruction prefix. We are particularly interested in whether this pure-synthetic approach can outperform, and eventually replace, the mainstream approaches based on human-originated signals [1], which forms the motivation of this work.
>
> Secondly, **Magpie serves as a strong baseline**. Models trained on Magpie have been reported to achieve state-of-the-art performance across all publicly available instruction-tuning datasets[2]. As such, the result that training with our datasets outperforms training with Magpie effectively positions our dataset as the best among all available options.
>
> We will make the motivation of comparing with Magpie more clear in the final manuscript.
>
> Additionally, we would also like to politely mention that in Section 4, **we have adopted another alternative dataset**, Ja-Self-Inst[3],  for Japanese experiments. Models trained on our dataset outperform those trained on Ja-Self-Inst.
>
> > Line 106 – Should this say “maximize log probability” instead of minimize? Line 212 – Should be “one of few” Line 315 – “Four” should not be capitalized.
>
> Thank you for pointing out the typos. You are absolutely right, and we will be sure to correct the typos in the final manuscript.
>
>
> [1] Self-Instruct: Aligning Language Models with Self-Generated Instructions. Wang et al. ACL 2023.
> [2] Magpie: Alignment Data Synthesis from Scratch by Prompting Aligned LLMs with Nothing. Xu et al. ICLR 2025.
> [3] Rapidly developing high-quality instruction data and evaluation benchmark for large language models with minimal human effort: A case study on japanese. Sun et al. LREC-COLING 2024.

---

> ### Author Response · Authors · 2025-06-03
> **We Are Open to Discussions**
>
> Dear Reviewer 7i5D,
>
> Again, thank you for taking the time to review our paper and for your positive and thoughtful feedback. As the final stage of the discussion period has begun, we would like to express our sincere interest in discussing with you. Below we have provided our responses, in the hope of addressing your concerns. We would greatly appreciate it if you could review the responses and let us know if they are clear enough. Also, please feel free to share any further concerns you may have. We look forward to your reply.

---

### Comment · Area_Chair_ke3k · 2025-06-05
**Reviewers, please engage**

Requesting the reviewers to read all reviews and responses and consider the arguments presented so that we can move closer to consensus on this paper.

Thank you!
Area Chair

---

> ### Author Response · Authors · 2025-06-06
> **Thank you for the reminder**
>
> Dear Area Chair,
>
> Thank you for keeping track of the discussion status of our manuscript!
>
>
> Dear reviewers,
>
> As mentioned by the Area Chair, we are awaiting your feedback. As for now, gratefully, **three out of four reviewers have provided positive ratings**, acknowledging our contributions in: (1) proposing data construction methods (7i5D, DRap, UvMA), (2) publishing language resources (UvMA), and (3) providing insights for instruction tuning in non-English languages (DRap, UvMA, WhHW). Dear reviewers 7i5D, DRap, UvMA -- thank you so much!
>
> Dear reviewer WhHW, we truly appreciate your comments as well. However, based on your feedback, it seems that there might be some potential misunderstandings. We have composed our response with the hope of clarifying these points and would be grateful to hear back from you. We are happy to continue the discussion if you have any further concerns. Thank you in advance for your time and patience.

---

### Decision · Program_Chairs · 2025-07-08

**Decision:**

Accept

**Comment:**

The paper investigates whether instruction-tuning datasets built from human-written instructions paired with LM-generated responses can outperform fully synthetic datasets like Magpie. Using real instructions from LMSYS-Chat-1M and responses from open-weight LMs (e.g., Llama-3.1, Gemma-2), the authors construct new datasets and demonstrate superior or comparable performance across multiple base models and in both English and Japanese.

Pros:
- The pipeline is simple, reproducible, and publicly available.
- The authors explore cross-lingual instruction tuning by translating instructions to Japanese, demonstrating generalization, and analyze the limits of tuning versus pretraining for handling cultural knowledge.
- The approach outperforms or matches state-of-the-art datasets like Magpie (a state of the art instruction tuning dataset) across multiple models.
- The approach uses real human-written instructions, improving instruction diversity and realism.
- The paper provides insightful analysis on instruction tuning vs. pretraining needs.

Cons:
- The work relies heavily on MT-Bench; lacks broader or human evaluations.
- The experiments focus on single-turn data, limiting applicability to multi-turn tasks.
- Japanese extension uses machine-translated instructions, which may introduce artifacts.
- The paper lacks direct comparison using synthetic instructions generated from the same seed data.
- Reviewers point out that some analyses (e.g., Japanese performance, LM judge reliability) could be deeper.

One reviewer (WhHW) identified a discrepancy between the paper's stated research claims and the experimental setup, requesting a more controlled experiment.  The meta-reviewer encourages the authors to more carefully scope the claims of the paper and, in the paper, identify the missing experiment as a possibility for future work.  Several reviewers asked for content to be moved into the paper (or appendix), and the authors are advised to take these recommendations seriously.